# EventNarrative: A Large-scale Event-centric Dataset for Knowledge Graph-to-Text Generation

**Anthony Colas** [*], **Ali Sadeghian**[*], **Yue Wang, Daisy Zhe Wang**
Department of Computer Science, University of Florida
{acolas1, asadeghian, yue.wang1, daisyw}@ufl.edu

## Abstract

We introduce EventNarrative, a knowledge graph-to-text dataset from publicly available open-world knowledge graphs. Given the recent advances in event-driven Information Extraction (IE), and that prior research on graph-to-text only focused on entity-driven KGs, this paper focuses on event-centric data. However, our data generation system can still be adapted to other types of KG data. Existing large-scale datasets in the graph-to-text area are non-parallel, meaning there is a large disconnect between the KGs and text. The datasets that have a paired KG and text, are small scale and manually generated or generated without a rich ontology, making the corresponding graphs sparse. Furthermore, these datasets contain many unlinked entities between their KG and text pairs. EventNarrative consists of approximately 230,000 graphs and their corresponding natural language text, six times larger than the current largest parallel dataset. It makes use of a rich ontology, all the KGs entities are linked to the text, and our manual annotations confirm a high data quality. Our aim is two-fold: to help break new ground in event-centric research where data is lacking and to give researchers a well-defined, large-scale dataset in order to better evaluate existing and future knowledge graph-to-text models. We also evaluate two types of baselines on EventNarrative: a graph-to-text specific model and two state-of-the-art language models, which previous work has shown to be adaptable to the knowledge graph-to-text domain.

## 1 Introduction

Natural language generation (NLG) is a rapidly developing area of natural language processing (NLP). With the advent of transformer-based language models, such as BERT [5], GPT-2 [35], XLNet [53], BART [21], UniLM [3] T-5 [36], and ERNIE [45], NLG has seen some recent advances in abstractive summarization, dialog response generation, and generative question answering. These tasks in NLG have all been accompanied by previously curated large-scale parallel datasets, where parallel denotes a tightly coupled input/output, allowing for generalized fine-tuning, including: the CNN/DM dataset [13, 28] and Gigaword [40] for abstractive summarization, Persona-Chat [54] and DSTC7 [7] for dialogue response generation, and CoQA [38] for generative question-answering. While the aforementioned NLG tasks have had a history of curated large-scale datasets to finetune on, the task of knowledge graph-to-text has been missing this scale of parallel data.

Knowledge graph-to-text generation is the process of taking structured data in the form of a knowledge graph (KG), which is a collection of subject-predicate-object $(s, p, o)$ triples, and describing the graph through natural language sentence(s). KGs describe real-world entities and their properties and tend to be incomplete. There are over 1,300 publicly available KGs with over 100B triples [14, 26], containing structured knowledge about biomedicine, geography, socioeconomic, life sciences, chemistry, publications, etc., and many cross-domain KGs. Some popular KGs include Wikidata [47], YAGO [37], ICEWS [31], and DBpedia [20]. These KGs are both user curated and extracted from

35th Conference on Neural Information Processing Systems (NeurIPS 2021) Track on Datasets and Benchmarks.

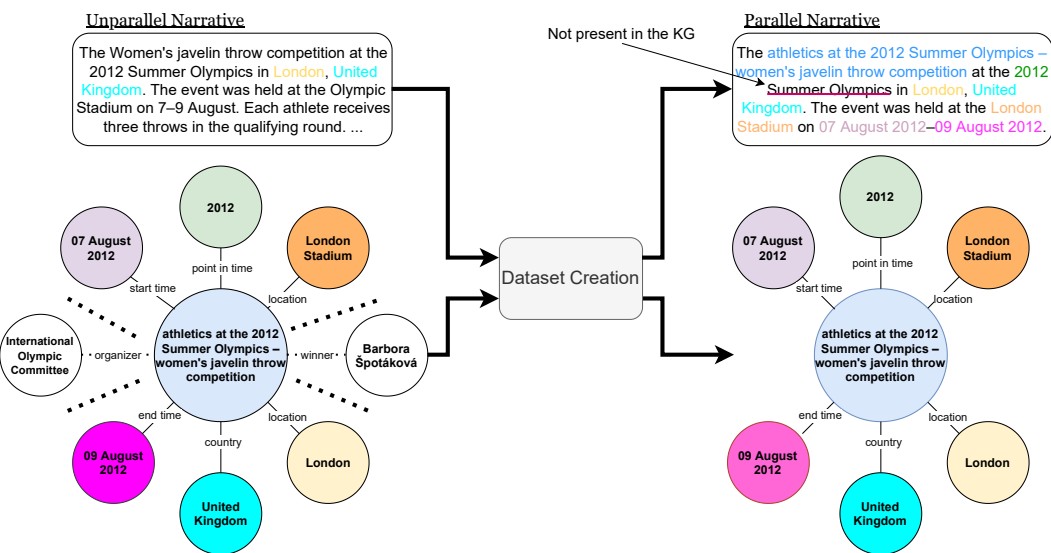

Figure 1: An example source and output narrative for the "Athletics at the 2012 Summer Olympics — Women's javelin throw" event. **Left:** The original nonparallel narrative with its corresponding KG from EventKG/Wikidata. Entities in the KG and text are linked by their mutual colors. **Right:** The filtered text and KG for the corresponding event. These are both output from our Dataset Creation approach. We highlight the corresponding entities in the texts and KGs.

Wikipedia text. Nevertheless, there are large disconnects between the data and natural language text. Resolving these disconnects, enables us to better serve the vast amount of structured information in the KGs in a user-friendly manner. One solution is finding methods to directly link the KG to its corresponding Wikipedia narrative when possible. We can then use the curated datasets to train NLP models to expand graph narrative generation to other KGs with fewer resources. Figure 1 illustrates an example Wikidata event graph with its corresponding narrative from Wikipedia.

The parallel datasets that currently exist for the knowledge graph-to-text generation are often small in size and do not take advantage of the KG ontology when creating the data. For example, the 2017 WebNLG Challenge dataset (WebNLG 2017) only contains 21,855 pairs of graphs and texts, while requiring an expensive hand-annotated operation to generate text on a given graph [8]. Another prominent parallel dataset, the AGENDA dataset, contains 40,000 examples [16] but is generated through the SciIE tool [25] on scientific article abstracts with only 7 different relations. This sparse ontology does not follow any standard KG ontology and induces sparse graphs paired with long texts [16]. Moreover, the dataset contains many entities that are isolated from their KG. We further discuss these and other knowledge graph-to-text datasets in Sections 4 and 2.

Current knowledge graph-to-text datasets are also entity-centric, containing data which are incompatible to narrate events. Events involve multiple actors, complex relations, various lengths, and temporal information, making them more information dense and variant, as detailed in Section 4. Numerous event-centric KGs [31, 19, 9] exist, which are valuable to narrate, and recent work has also looked into how to best extract events from text [4, 6, 22]. We therefore develop a more comprehensive algorithm that matches event-centric KGs to their natural-language narration. Many similar events occur frequently but at different times. Thus, our entity matching algorithm has a date matching component to ensure the KG-text pairs contain date/time information. For example, there is a "2014 FIFA World Cup" and "2018 FIFA World Cup", where the event descriptions may have high overlap. We refer to each text instance as a *narrative* of the graph/event, as when describing an event, one is often described to be *narrating* the event [30].

Events are distinct in their length, occurrences, properties, and relations involved. Our dataset reflects this, containing events from different time periods, having thousands of types, and containing approximately 650,000 triples. Therefore, EventNarrative overcomes various shortcomings of existing datasets: it is approximately 6 times larger than the current largest parallel dataset, knowledge graph-text pairs are generated automatically via an existing rich ontology, and there are over 7,000 different

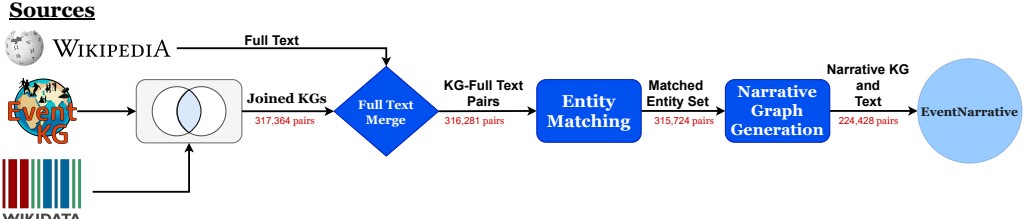

Figure 2: An overview of our data creation approach.

types of events, ranging from sports seasons to social media campaigns. Relations within EventKG include location, event type and start/end time.

We propose EventNarrative, a large-scale supervised graph-to-text dataset, used to facilitate research in event-based conditional text generation. EventNarrative is extracted and paired from existing large-scale data repositories, including Wikidata, Wikipedia, and EventKG [9]. EventKG is a multilingual event-centric temporal KG, which combines event graphs from Wikidata, DBpedia, and YAGO. We begin by first extracting events from EventKG, and then for each event, augment the data with additional corresponding Wikidata information. In total, EventNarrative contains approximately 220,000 data pairs.

We establish baselines on EventNarrative by comparing current state-of-the-art knowledge graph-to-text models on our automatically extracted test set. Future versions of the EventNarrative dataset will aim to enrich the current dataset by incorporating other KGs such as DBpedia and YAGO.

Altogether, our contributions are as follows:

- A large-scale, event-centric, parallel knowledge graph-to-text dataset of approximately 225,000 KG-text pairs, spanning over 7,000 event types, and over 650,000 triples.
- A comprehensive entity matching and knowledge graph-to-text matching algorithm that automatically pairs KGs to natural language texts.
- Benchmark evaluations and baselines results on EventNarrative.

Our dataset can be found on: `https://www.kaggle.com/acolas1/eventnarration`.

## 2  Related Datasets

One of the early knowledge graph-to-text datasets, WebNLG 2017 [8], is a human-annotated parallel dataset that consists of 27,731 graph/text pairs and 9,674 unique graph instances, therefore containing multiple text samples per graph. After the WebNLG 2017 challenge, the dataset was expanded from 9 to 15 categories. As shown in table 1, each KG was extracted from DBpedia. Because the dataset is handcrafted, there is a high precision between the matches in the KGs and text, but the dataset does not scale.

The AGENDA dataset, first introduced by Koncel et al. [16] was constructed by first collecting approximately 40,000 scientific articles from Semantic Scholar, then extracting KGs from the text using the SciIE [25] tool. The dataset contains only 7 relations and many entities which are not connected to a KG. Moreover, the dataset is not bound to any ontology, making it difficult to expand the KG component of the dataset in a standardized fashion.

A recent established non-parallel dataset, GenWiki, was constructed by matching Wikipedia articles with DBpedia entities [15]. However, its goal is to develop a large-scale, non-parallel dataset for unsupervised graph-to-text learning, where all elements in the graphs are not necessarily contained in the text. GenWiki also has a third component, namely entities, which are extracted from the text, but not necessarily contained within the graph. These entities are used to construct both the graph and text for both the graph generation and text generation tasks, respectively. While valuable for unsupervised learning in knowledge graph-to-text, we note that this may not model the knowledge graph-to-text problem, where all entities should be contained within the KG and one may not have access to the text in order to extract the shared entities. Similarly, WikiGraphs was created by matching

WikiText-103 [27] articles with KGs from Freebase [48]. While the KGs and text in WikiGraphs are not tightly coupled, WikiGraphs instead focuses on pairing large KGs, containing about 39 nodes per KG, with large texts (complete Wikipedia articles). While valuable for generating long text, this may not accurately represent the supervised graph-to-text problem, where the text should be a complete representation of the graph. Thus, WikiGraphs is most comparable to works such as GenWiki [15].

EventNarrative sources event items from the more recently established EventKG [9], an event-centric fusion-based KG. Previous works employing EventKG include work on timeline generation [10], event series completion [11], and event-centric question answering [44]. While EventKG contains facts which involve location, time, and event actors, we enrich EventNarrative by extracting all related facts from a given event item, so that the graphs can better align with their corresponding narratives. Although out of scope for this paper, it is worth mentioning some related work on KG completion and temporal prediction that only rely on the KG itself [24, 43, 17, 42, 1, 41, 29].

# 3 Dataset Creation

This section explains our data creation process. We detail the various data sources used to extract events, our algorithm to match entities and text, and the graph generation technique we used to produce our final narratives and KGs. An overview of our methodology can be found in Figure 2.

## 3.1 Sources

The knowledge graphs in EventNarrative are first sourced from EventKG [9], a multilingual event-centric KG which incorporates events from Wikidata, DBpedia, YAGO, the Wikipedia Current Events Portal, and the Wikipedia events list. Each event contains information on where it was extracted from and any aliases or alternative names for the event, if any. Properties of these events in EventKG include temporal information such as the *hasBeginTimeStamp*, *hasEndTimeStamp*, *startUnitType*, and *endUnitType* relations, as well as spatial information with *hasPlace*. Events are also connected with the *previousEvent* and *nextEvent* relations. We then filter and keep events which are extracted from Wikidata and linked to a Wikipedia page. We do so because Wikipedia articles, and thus the items found in the text/narrative, are directly used to construct Wikidata, which can be queried through the Wikidata query service using an article's Wikidata Q identifier (QID). In total, we collect an initial set of *322,674* graph events that have both a Wikidata and Wikipedia resource link through EventKG.

EventKG contains a large number of events, with information pertaining to location, time, as well as actors involved in the events. Yet, many relations and properties found on Wikidata can be missing from EventKG. We therefore query Wikidata to get an event item's related properties, objects, and labels through the SPARQL-based Wikidata query service[1]. We do so by obtaining an event's Wikidata QID from EventKG and querying the Wikidata online query service, taking precautions to not overload the service by writing aggregate queries to obtain results for multiple events in parallel. We also query for each object's QID, as it is of use in the entity matching component. Because some relations and properties between EventKG and Wikidata overlap, we first normalize all temporal or location-based relations from EventKG into their corresponding Wikidata property names and remove any duplicates. Next, we filter any events which do not contain a type, e.g., sports season, battle, or election. This type is represented though the relation *wikidata_type_label* from EventKG or *instance of* from Wikidata. By keeping types, future research can filter EventNarrative by event type or incorporate the types into their knowledge graph-to-text models. After filtering, the dataset contains *317,364* graph events.

When narrating an event, important details such as actors involved or sub-events often lie deeper inside the Wikipedia article itself. While previous work on both table-to-text and knowledge graph-to-text has limited the size and locality of the textual data [18, 50, 15], we retrieve an event article's whole Wikipedia text to capture all of an event's textual details—the Full Text Merge step. We then filter out extraneous details contained within square or curly braces, which typically denote altered or omitted information. Consequently, some Wikidata events have no Wikipedia article. After retrieving the whole text, there are *316,281* KG-text pairs.

---

[1]https://www.mediawiki.org/wiki/Wikidata_Query_Service

## 3.2 Entity Matching

To match the in-text tokens with their corresponding KG triples, we devise an extensive entity matching technique which is specialized for event data, but can be fitted to capture other types of data. First, similar to Wang et al. [50], we locate all the hyperlinks in the Wikipedia text and identify their QIDs. After doing so, we match these QIDs with the ones captured from the Wikidata graph in the previous step, preserving those entities which have a match. In-text entities in Wikipedia articles often do not have any links to Wikidata. To overcome this, we check for exact matches between the Wikidata property items and in-text tokens, saving those with a match. One drawback of the exact match method is that Wikidata entities are shorter in length and may overlap with Wikipedia text that was better suited for a longer Wikidata entity. We therefore first sort all the Wikidata property entities by length, longest to shortest, before executing our exact match step. This step also allows us to match properties from the Wikidata graphs that are numerical or dates, but only if the date format matches that found within Wikipedia text.

**Dates** are crucial components for any event-centric dataset. We therefore design a separate module to match Wikidata dates within our KG set to those from the Wikipedia text, making our entity matching algorithm biased towards event-centric data. In order to construct an exhaustive search algorithm for in-text Wikipedia dates, we refer to the Wikipedia Style Manual [2] which defines acceptable date formats when editing Wikipedia articles. We write regular expression patterns (RegEx) to match these formats as well as others observed when manually reviewing the EventNarrative dataset. By manually iterating through both the Wikidata graphs and Wikipedia text, we note that many dates only have an overlap in month or year. Therefore, if a date from Wikidata is not found through the RegEx patterns, we match the date if it contains a monthly or yearly overlap. We replace those matches in the narrative with their corresponding match in the KG in order to normalize the graph-text pairs. After performing the entity matching step, we filter out those pairs for which we found no matches, keeping *315,724* KG-text pairs.

Our entity matching technique prioritizes a high recall, where dates and entities may overlap but not correctly match. To verify our entity matching technique, we sampled 500 events from our final dataset and recruited workers to note any errors in the data related to the linked entities and matching relations. Details can be found in Section 4.

## 3.3 Narrative KG Generation

For every graph-text pair $(G, T)$ obtained so far, we recursively discard sentences (and nodes) from $T$ (and $G$) until all remaining sentences have at least 2 entities in $G'$. This is necessary in order to reduce textual noise. Otherwise, we would have texts that could not be generated given the information contained in the graph. Figure 1, shows an example parallel narrative, which contains triples from the KG. During this process, if any of the $G$ or $T$ become empty, we discard the pair. The result is a connected graph which represents the current filtered narrative text. One could achieve similar results using a simpler sentence level approach. However, this rather intricate approach of constructing the graph-text pairs using the complete text preserves triples that are not fully explainable through a single sentence. In the end, we are left with **224,428** KG-text pairs. More in-depth analysis on the final EventNarrative dataset are presented in Section 4.

## 3.4 Limitations

Our dataset construction methodology is not without limitations. The narrative KGs are of course limited to the elements and relations found in Wikidata, which itself is incomplete, causing it to miss entities found within the narrative text. We knowingly discard sentences which may contain co-references to events. While initially creating the dataset, we experimented with current state-of-the-art co-reference resolution systems, but they performed poorly on event-centric data. This may be because of the overlap between event names and their property names, e.g., "2013-2014 Manchester City F.C. season" and "Manchester City". Because event names and their properties may have a high overlap, we do not perform any fuzzy matching techniques to extract events.

---

[2]https://en.wikipedia.org/wiki/Wikipedia:Manual_of_Style/Dates_and_numbers#Formats

Table 1: Overview of current knowledge graph-to-text datasets. A dataset has an Entity Match if all entities from the KG are contained in the narrative. A dataset has a Triple Match if all entities are linked to the KG's triples.

| Dataset | KGs | Entity Match | Triple Match | Domain | Ontology | Text | Parallel |
|---|---|---|---|---|---|---|---|
| WebNLG 2017 | 9,674 | ✓ | ✓ | 15 Categories | DBpedia | Crowdsourced | ✓ |
| AGENDA | 40,720 | ✓ | ✗ | Semantic Scholar | N/A | Scientific abstracts | ✓ |
| GenWIKI (full) | 1,336,766 | ✗ | ✗ | General Domain | DBpedia | 1-10 Wiki sentences | ✗ |
| GenWIKI (fine) | 757,152 | ✗ | ✗ | General Domain | DBpedia | 1-10 Wiki sentences | ✗ |
| WikiGraphs | 23,522 | ✗ | ✗ | General Domain | Freebase | Full Wiki text | ✗ |
| EventNarrative | 224,428 | ✓ | ✓ | Events | Wikidata | Full Wiki text | ✓ |

## 4 Dataset Analysis

We compare EventNarrative with four popular knowledge graph-to-text datasets, including: WebNLG 2017 [8], AGENDA [16], GenWiki [15], and WikiGraphs [48]. Note that because WebNLG 2017 contains multiple texts per graph with an n:1 relationship, we decouple each text from its corresponding graph before analyzing the data. Though GenWiki is a non-parallel dataset primarily constructed for unsupervised learning, we wish to also highlight other key differences. We include both renditions of GenWiki, *full* and *fine*, where *fine* has a tighter entity overlap threshold. Likewise, we include WikiGraphs, though each KG is loosely coupled with their corresponding Wikipedia text.

### 4.1 Dataset Synopsis

We begin by first performing a high-level analysis of current knowledge graph-to-text datasets. Among all the parallel datasets in Table 1, our proposed EventNarrative is the largest, having 6 times more KGs than the second largest dataset (AGENDA) and 25 times more than the manually annotated WebNLG. Although GenWiki is larger than EventNarrative, many entities and relations from the triples are not contained within the text and many entities from the text are not found in the graphs. The dataset is purposely created for unsupervised learning and does not model the knowledge graph-to-text supervised task. As AGENDA contains isolated entities that do not belong to triples, the only other dataset with perfect matches between the text and graphs, WebNLG 2017, was hand-crafted, limiting the number of samples generated. Since the text was handcrafted in WebNLG-2017, iterative improvements on the dataset such as standardizing the entities and relations within the text based on an ontology becomes extremely challenging. Conversely, because the narratives found in EventNarrative are sourced from Wikipedia, various existing tools can be used to improve the entity matching algorithm. Therefore, EventNarrative is the only dataset that is large-scale, ontology-based, utilizes an open real-world KG, may be used for supervised learning, and contains connected graphs that are fully contained within a text narrative.

### 4.2 Statistical Analysis

We now take a closer look at EventNarrative, demonstrating that our dataset contains a large amount of variable data, with closely aligned KG-narrative pairs. Table 2 presents some in-depth statistics between the current supervised knowledge graph-to-text datasets, including: WebNLG 2017, AGENDA, and the proposed EventNarrative. We exclude GenWiki and WikiGraphs from this analysis because these datasets does not meet the requirements of being parallel, and their entities/triples do not completely align with the text.

From table 2 we see that EventNarrative has approximately 110 times more entities, 2 times more relations, and 8 times more triples than WebNLG 2017, the only other dataset containing no disconnected entities between the KG and text. Unlike [15], we choose not to limit our relations for two reasons: (1) our dataset is not open-domain as in [15], (2) our aim is to build a challenging dataset which simulates real-world KGs. Even so, with 305,685 entities and 224,428 graph-narrative pairs, 672 relations are tractable.

While the AGENDA dataset contains more triples on average per sample, it contains less than one triple on average per sentence, while also holding the longest mean text length throughout all of

---

[2]Note that we compare the WebNLG 2017 dataset released after the challenge, which contains 15 categories found on: `https://webnlg-challenge.loria.fr/challenge_2017/`.

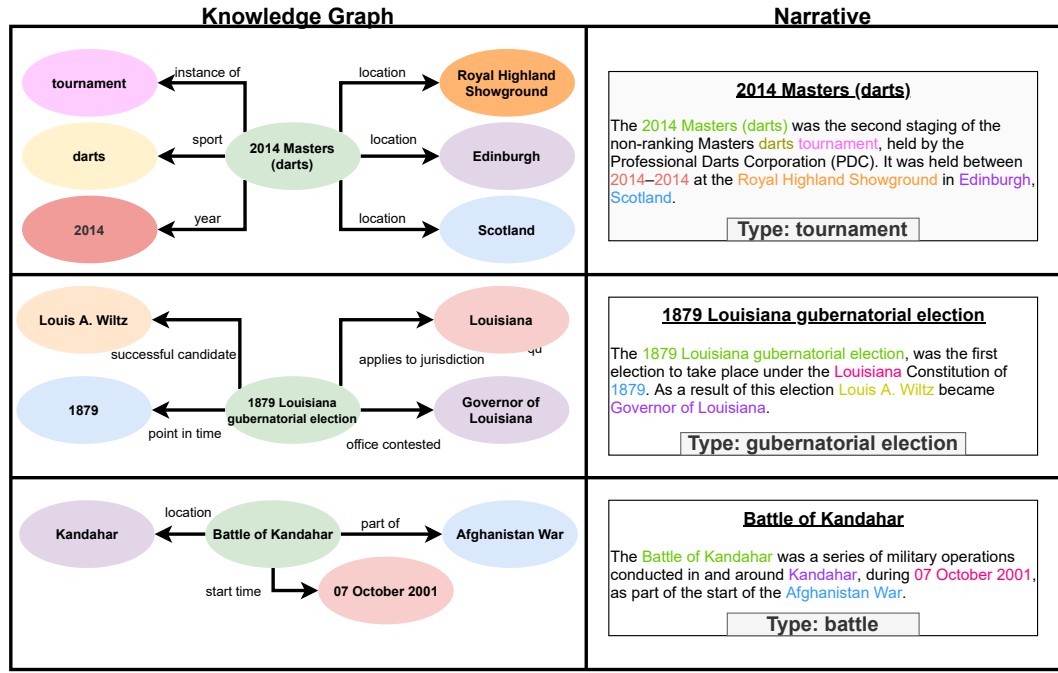

Figure 3: Examples of KG-narrative pairs from EventNarrative. Linked entities are color coded.

Table 2: Statistics of the current parallel knowledge graph-to-text datasets. We report the total number of KG components, number of tokens in the narratives (including the percentage which encompass entity tokens), and overall mean statistics for both the KGs and narratives.

| Dataset | KG | | | Tokens | | | Mean | | | |
|---|---|---|---|---|---|---|---|---|---|---|
| | Entities | Relations | Triples | Total | Unique | Entity | Triples | Text Length | Entities in Text | Triples/Sentence |
| WebNLG 2017 | 2730 | 354 | 81,927 | 623,902 | 8,075 | 60% | 2.95±1.55 | 22.50±11.24 | 57%±23 | 2.12±0.98 |
| AGENDA | 159,691 | 7 | 180,603 | 5,760,660 | 77,896 | 27% | **4.43±3.19** | **141.47±40.92** | 27%±7% | 0.81±0.74 |
| EventNarrative | **305,685** | **672** | **656,302** | **11,352,387** | **222,338** | **25%** | 2.92±2.48 | 50.58±58.59 | **32%±13%** | 1.82±1.12 |

its samples [3]. EventNarrative contains an average of about two triples per sentence, and contains a more stable text length at approximately 51 tokens. Though both the AGENDA and EventNarrative datasets are almost equivalent in the percentage of text tokens that are entities, recall that in AGENDA not all entities belong to a triple, with approximately 49% of the entities missing from their KG.

As expected, WebNLG 2017 has the highest percentage of entity tokens in the overall text, average percentage of entity tokens in its text per sample, and average number of triples per sentence, as the dataset is hand-crafted and human verified. Though automatically generated, EventNarrative is comparable to WebNLG 2018 in number of triples per sentence, suggesting that our dataset creation process may closely simulate human annotation.

Another notable aspect of EventNarrative is its high variability between samples, as shown by the variance in the number of triples and text length. This makes EventNarrative a complex yet practical dataset for modeling the knowledge graph-to-text problem, verifying our claim that event data is highly variable in length.

We illustrate the distributions of event types and relations in figure 4. The top 10 event types include: sports season, American football season, sporting event, tennis tournament edition, battle, election, basketball team season, association football team season, Olympic sporting event, and nation at sport competition. The top 10 relations include: point in time, location, country, sport, start time, instance of, end time, part of, sports season of league or competition, and winner. While EventNarrative is highly variable in its length per sample, the type and relation distributions reveal that a plurality of the events are sports related. Intuitively, this makes sense, as there are many yearly and even monthly

---

[3] All data were tokenized with NLTK: https://www.nltk.org/

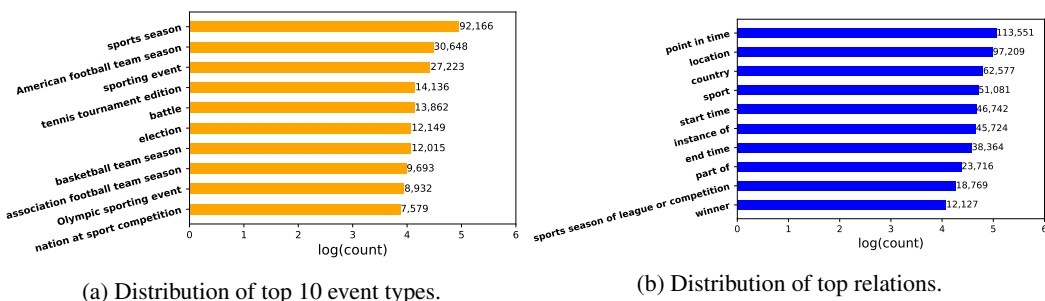

(a) Distribution of top 10 event types.  (b) Distribution of top relations.

Figure 4: Distribution of most frequent types and relations in EventNarrative.

Table 3: Qualitative analysis results: entities correct (EC), entities incorrect (EI), relations correct (RC), relations incorrect (RI).

(a) Correct/incorrect per KG-narrative pair.

|     | EC  | EI  |
| --- | --- | --- |
| RC  | 397 | 57  |
| RI  | 30  | 16  |

(b) Total correct entities and relations (percentage).

| % CE          | % RC          |
| ------------- | ------------- |
| $95.56 \pm 4.0$ | $95.50 \pm 4.0$ |

sports-related events. EventNarrative also includes a substantial amount of battles, award ceremonies, legal cases, finals, bilateral relations, and elections. Future work using GraphWriter can filter events by types based on their needs.

### 4.3 Qualitative Analysis

In this section, we evaluate our dataset based on how closely it resembles human annotations. To do so, we recruit 3 annotators to manually check 500 randomly sampled KG-text pairs from EventNarrative. To ensure quality, we recruited annotators that have experience in KG research. We first ask the annotators to verify (and count) if all entities and relations contained within the KG are correct with respect to the narrative. We do so in order to verify both the Entity Matching and Narrative KG Generation steps from our dataset creation process. Table 3 demonstrates that both the entities and relations from the KGs are indeed tightly coupled with their respective narratives. When aggregating all the annotators' results, approximately 96% of all entities and 95% of all relations in the sample were deemed correct. On a coarser level, 397 of the KGs had no errors of any type. We found an 100% agreement when measuring the kappa value among the annotators, meaning that all annotators found entity, relation, or both types of errors in the same samples. To more closely evaluate each data generation step, including the sources and RegEx matching, we asked annotators about the different errors they encountered. They reported that most entities which are incorrectly paired are either mischaracterized by Wikipedia or are dates that have different scopes between the KG and source narrative; i.e., containing the day and month of an event within the text but only containing the year within the KG.

Three different types of events from EventNarrative are shown in figure 3: tournament, gubernatorial election, and battle. The first row shows an example error in our date processing. The original text stated "1–2 November 2014" but is incorrectly replaced with "2014-2014" because of the different scope of dates between the KG and narrative.

## 5 Benchmark Evaluation

Our aim is to establish a dataset which can be used to help advance the state-of-the-art in transforming knowledge graphs to natural language narratives. We begin this work by comparing established supervised knowledge graph-to-text baselines on EventNarrative. Currently, there are two approaches to this task: first, modeling the problem with a graph transformer-based network; second, treating the problem as a summarization task by finetuning on pretrained language models (PLMs). The graph

Table 4: Narrative generation results for EventNarrative. The best result for each metric is in bold.

| | BLEU | chrF++ | CIDEr | METEOR | ROUGE | BERTScore |
|---|---|---|---|---|---|---|
| GraphWriter | 30.78 | 47.91 | **4.59** | **27.72** | **71.92** | 92.12 |
| T5$_{base}$ | 12.8 | 56.76 | 3.00 | 22.77 | 52.06 | 89.59 |
| BART$_{base}$ | **31.38** | **64.71** | 3.31 | 26.68 | 62.65 | **93.12** |

transformer-based network we experiment with is GraphWriter [16], a model which can capture local and global information when encoding a graph. Second, we experiment by finetuning on two prominent pretrained language models (PLM), BART [21] and T5 [36], which have been shown to outperform graph-to-text specific models on the AGENDA and WebNLG 2017 datasets [39].

## 5.1 Experimental Setup

We divide the dataset into an 80/10/10 train/dev/test split. During the training of BART and T5, we frequently evaluate on the dev set for model selection. Due to the high computational overhead imposed by the decoding step, we chose to only use a subset of the dev set for training. We use this same random subset for model selection for both BART and T5. For all models, the final reported metrics are on the full test set. All experiments were performed on NVIDIA RTX 2080 Ti GPUs.

For the GraphWriter model, we use the version provided by Guo et al. [12] which utilizes the Deep Graph Library (DGL) [49]. We keep its default parameters of: a learning rate of $2 \cdot 10^{-4}$ size batch size of 32, a beam size of 5, 4 attention heads, and train for 30 epochs[4].

To finetune on EventNarrative with BART and T5, we follow a procedure similar to that of [39], prepending "translate from Graph to Text" to the source (graph) data for the T5 model and adding the subject $$, predicate $<P>$, object $<O>$ tokens into the vocabulary for both PLMs. All of our PLM experiments are done using the base models released by HuggingFace [52]. Given the computational complexity of BART and T5, we choose to follow [39]'s setup, using the Adam optimizer and linearly decreasing learning rate scheduler without warm-up with an initial learning rate of $3 \cdot 10^{-5}$. We use a batch size of 2 and beam search size of 3. The dev set's BLEU score is used for model selection.

## 5.2 Results

We evaluate EventNarrative on frequently used NLG evaluation metrics: BLEU [32], chrF++ [33, 34], CIDEr [46], METEOR [2], and ROUGE [23]. Additionally, as in [39], we also evaluate the test set using BERTScore [55] which computes text similarity based on contextualized embeddings. Table 4 presents the results for each baseline model. Overall, for the EventNarrative dataset, the GraphWriter and BART models give similar results, both significantly outperforming T5. The GraphWriter model outperforms BART on CIDEr, METEOR, and ROUGE_L, while BART performs best on BLEU and BERTScore. The poor results of T5 may be because of the difference in data that BART and T5 were originally trained on. BART was trained on news articles, which can closely resemble events. Overall, our results show that graph-to-text specific models are still competitive to PLMs and deserve further investigation.

## 6 Discussion and Conclusion

EventNarrative closely resembles the available KGs and can be used to generate narratives in a supervised manner. This is enabled by its large size, rich ontology, and variability within the data. Our human qualitative analysis verified that about 96% of entities and relations are correctly matched.

Our dataset generation framework is automated, therefore EventNarrative can be re-assembled and extended with other ontological KGs such as DBpedia or YAGO. We will periodically improve and update EventNarrative, as the nature of the dataset depends on continuously adding new events. The dataset generation framework can also be adapted to other types of entity-centric data in order to generate rich and more tightly-coupled sets of knowledge graph-to-text data.

---

[4]For more details, see `https://github.com/QipengGuo/CycleGT`

EventNarrative provides the community with new challenges because of the variety within its data, while also providing new insights into knowledge graph-to-text baselines. Previous parallel datasets have been lacking because of their size, loosely coupled triples, and sparsity, all of which can saturate the results of current baselines. EventNarrative is tightly coupled, allowing researchers to focus on generating proficient models which narrate real-world KGs. We hope that EventNarrative can enable ground-breaking new work in knowledge graph-to-text and event-centric research.

## 7 Broader Impact

EventNarrative can assist researchers in other fields in studying different graph structured data (e.g., events that are represented as graphs) by providing them with easily readable narratives. At the same time, as a text generation dataset, there are risks concerning generating fake news and disinformation, specifically related to recent or current events which may appear in the dataset. This can especially occur if the language produced from the KG looks fluent but is completely fabricated [51]. While all of our narratives are extracted from Wikipedia and this issue may not be apparent, we discourage anyone from substituting any text with those that may spread disinformation.

## Acknowledgements

This work is partially funded and supported by the GSPA at the University of Florida, the McKnight Doctoral Fellowship, the NSF under IIS Award #1526753, and DARPA under Award #FA8750-18-2-0014(AIDA/GAIA). We would also like to thank the annotators and members of the Data Science Research Lab at the University of Florida who helped throughout this work.

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
