# OpenReview forum: "EventNarrative: A Large-scale Event-centric Dataset for Knowledge Graph-to-Text Generation"
_NeurIPS.cc/2021/Track/Datasets_and_Benchmarks/Round1 — NeurIPS 2021 Datasets and Benchmarks Track (Round 1)_

### Official Review · Reviewer_APbv · 2021-06-23
**Exciting new dataset, needs clarification about baseline evaluation**

**Rating:** 7
**Confidence:** 5

**Strengths:**

- large-scale
- event-centric
- parallel (filtered to guarantee good matches between texts and KGs)
- explicit links between graph nodes and entity strings

**Weaknesses:**

The paper does not evaluate a lot of baselines and their results could be discussed in more detail.
I also have some doubts about the way their experiments are described. Cf. below 'Correctness'.

after rebuttal: most of these weaknesses are explained and so the paper only needs to incorporate these clarifications.

**Additional Feedback:**

Your section 5 describes the existing datasets rather than compares them to your existing work. So I recommend putting it nearer to the beginning because it provides context for readers that are less familiar with existing datasets. After reading your analysis in section 3, I found that section 5 does not provide much more insight. A different order might work better.

As one of the filter criterions was that each sentence must contain at least two graph entities, I wonder if something like a serialization plan could be constructed for the text-graph pairs, i.e., the order of the triples as they appear in the text or some correspondence between sentences and triples. Recent work [2] has shown that this kind of annotations can be very useful.

Typo in L122: knowledge KG -> KG


[2] https://www.aclweb.org/anthology/2020.acl-main.224

**Clarity:**

L157-158, L193: your graphs are certainly not all "fully connected". You proably mean just "connected", right? "Fully connected" means that every node is connected to every other node. You mean that you don't have disconnected graphs, as I understand it.

**Correctness:**

The data collection seems reasonable and correct.

Regarding the baseline evaluation, I understand that BART and T5 are evaluated on a random subset of the dev portion. But the whole dev portion is used for model selection? That is a weird choice. Or do you mean that you evaluate only on a random subset of the test split after selecting your model on the dev split?
Also, are the results in Table 4 comparable? Is GraphWriter evaluated on the same exact subset of the data as BART and T5? Also, is the same random set of data used to evaluate T5 and BART or do you sample different subsets here?
If the data splits were not the same, the evaluation is a lot less meaningful.

**Documentation:**

Kaggle URL is given, with license etc.
In my opinion, the description in the paper is detailed enough to support reproducibility.

**Ethics:**

I do not have ethical concerns.

**Relation To Prior Work:**

The work offers a detailed analysis and comparison with existing popular graph-to-text generation benchmarks.

The work would benefit from mentioning the contemporaneous work "WikiGraphs"[1].


[1] https://www.aclweb.org/anthology/2021.textgraphs-1.7

**Summary And Contributions:**

This paper provides a new KG-to-text benchmark based on events and their narratives. It is larger than existing similar benchmarks, thus facilitating supervised learning of large neural models.
By the means of various filtering methods, the authors ensure a close correspondence between the text and the KG, i.e., the text is guaranteed to be predictable from the knowledge stored in the graph. Starting from human-authored Wikipedia text, the text, however, becomes more artificial by replacing all entities with their canonical strings from the graph. In some cases, this even leads to unnatural text, when, e.g., a year is present in the KG but the text contained a full date. The result then has phrases like "between 2014 and 2014". These limits are discussed and, based on a human evaluation, these errors are rare.
The event-centricity of the new benchmark is also a useful contribution that previous benchmarks lack although this type of formal knowledge is particularly suitable to be written out as text.

(modified: 20 Jul 2021)

---

> ### Author Response · Authors · 2021-07-10
> **Thank you for the constructive feedback and questions**
>
> We sincerely appreciate the detail that went into your review. Thank you for all of the constructive comments. For ease of reading, we have noted the comments in block quotes and their responses below:
>
> >Regarding the baseline evaluation, I understand that BART and T5 are evaluated on a random subset of the dev portion. But the whole dev portion is used for model selection? That is a weird choice. Or do you mean that you evaluate only on a random subset of the test split after selecting your model on the dev split? Also, are the results in Table 4 comparable? Is GraphWriter evaluated on the same exact subset of the data as BART and T5? Also, is the same random set of data used to evaluate T5 and BART or do you sample different subsets here? If the data splits were not the same, the evaluation is a lot less meaningful.
>
> We understand that our explanation in the paper might be unclear. We will add the following to the paper:
> “During of training BART and T5, we frequently evaluate on the dev set (for model selection) and due to the high computational overhead imposed by the decoding step, we chose to only use a sample of the dev set for faster training. We use this same random subset for model selection for both BART and T5. However, for all models, the final reported metrics are on the full test set. “
>
> >L157-158, L193: your graphs are certainly not all "fully connected". You proably mean just "connected", right? "Fully connected" means that every node is connected to every other node. You mean that you don't have disconnected graphs, as I understand it.
> Thank you for pointing out this typo. We have made the proper corrections.
>
> >The work would benefit from mentioning the contemporaneous work "WikiGraphs"[1].
> [1] https://www.aclweb.org/anthology/2021.textgraphs-1.7
>
> Thank you for bringing this paper to our attention. This work is a highly relevant and beneficial addition to our paper. We have mentioned the paper in our related datasets section and are waiting for their github page [Page not found · GitHub](https://github.com/deepmind/deepmind-research/tree/master/wikigraphs) to go public so we can perform a more in-depth comparison to include in our table 2.
>
> >Your section 5 describes the existing datasets rather than compares them to your existing work. So I recommend putting it nearer to the beginning because it provides context for readers that are less familiar with existing datasets. After reading your analysis in section 3, I found that section 5 does not provide much more insight. A different order might work better.
> Thank you, we agree that putting it nearer to the beginning would provide better context for readers, therefore we have moved Section 5 to Section 2.
>
> >As one of the filter criterions was that each sentence must contain at least two graph entities, I wonder if something like a serialization plan could be constructed for the text-graph pairs, i.e., the order of the triples as they appear in the text or some correspondence between sentences and triples. Recent work [2] has shown that this kind of annotations can be very useful.
> [2] https://www.aclweb.org/anthology/2020.acl-main.224
>
> Thank you for bringing this paper to our attention too. We believe this is an idea worth experimenting, but considering the necessary adaptations that we have to make and the time constraints, we will refer to the paper but leave the experiment for future works.
>
> >Typo in L122: knowledge KG -> KG
>
> Thank you for noting this typo. We have now made the correction.

---

> > ### Comment · Reviewer_APbv · 2021-07-13
> > **Thank you for addressing my questions and one clarification**
> >
> > Thank you for addressing all of my questions and doubts. You put them at ease.
> >
> > I want to clarify my comment regarding the order of triples or serialization plan. I did not mean to suggest that you should investigate if and how such a plan can improve performance on your benchmark. I agree that conducting further experiments would be too time-consuming.
> > My question was rather about providing additional annotations for the data. How difficult would it be to add an (automatic) annotation of the order, in which the triples are represented in the target text? How reliable would such an automatic annotation be? Is it possible to get a fully reliable ground truth for the order of triples by aligning the at least two entities mentioned in each sentence with the two entities mentioned in each triple? What challenges might the creation of this annotation face? If it is simple, would you be willing to provide this additional annotation?

---

> > > ### Author Response · Authors · 2021-07-13
> > > **Response to Serialization Question**
> > >
> > > Thank you very much for the clarification. We believe it should be fairly straightforward to automatically annotate the order in which the triples are represented in the text. We have designed our dataset so that there is a direct correspondence between the graph entities and entities in the text. Therefore, such annotation should be reliable as long as all occurrences of the entities (from the graphs) in the text are accounted for, e.g. one challenge would be that the text may contain an entity from the graph more than once. Thank you for this suggestion, and we will definitely keep this in mind when providing the first update of our dataset. Additionally, we hope this better addresses your comment.

---

> > > > ### Comment · Reviewer_APbv · 2021-07-14
> > > > **Follow-up suggestion about the serialization plan annotation**
> > > >
> > > > Thank you, this indeed addresses my comment better.
> > > > I see how it can be a challenge if more than one relation holds between the same two entities. Then, there would surely be more than one sentence mentioning these two entities together, too, and the alignment of triples and sentences would be ambiguous. I don't suppose you have collected any statistics about how often these cases occur?
> > > >
> > > > I see how it can be difficult to obtain these annotations automatically but it would still be interesting to know how often we can obtain the mapping between sentences and triples unambiguously. It would also be a nice way to quantify one aspect of the correspondence between graphs and texts you aimed for when designing your dataset.
> > > > From my point of view, it is not a strict requirement to add such statistics but the tight correspondence between graphs and texts is an interesting property of the data you collected and it would thus make the paper stronger to discuss it more.

---

> > > > > ### Author Response · Authors · 2021-07-15
> > > > > **Serialization Plan Annotation Reply**
> > > > >
> > > > > Since we have not instructed our annotators to control for this ambiguity, we do not think we are able to accurately estimate how often the mapping is unambiguous. We will take note of this during our first update of the dataset when giving instructions to our annotators. Again, thank you for bringing this point to our attention, as we agree that this is an interesting facet of the data.

---

### Official Review · Reviewer_kKsU · 2021-06-27
**A large-scale dataset for event-centric knowledge graph to text generation**

**Rating:** 7
**Confidence:** 3

**Strengths:**

The EventNarrative dataset created by the authors is claimed to be the largest dataset for event-centric KG to text generation. The dataset is created based on existing well established data sources e.g., EventKG, Wikidata, and Wikipedia. In this sense, it is a good contribution to the research problem of event-centric KG to text generation.

Because of the dataset is created based on easily accessible and highly accountable sources, there is no major concern on the raw quality of the data sources.



**Weaknesses:**

Authors lists a few limitations in Section 2.4 on the dataset itself mainly because of the creation process and the ignorance of co-reference resolution.  Other than the limitation on the dataset creation itself, it is not clear the unique characteristics of event-centric KG compared to generic KB with respect to the task of KG to text generation. The authors may want to provide a stronger justification on why such a dataset is essential to study KG to text generation.

Another weakness is on the quality of the dataset creation. The authors evaluated 500 randomly sampled KG-text pairs and reported a very good score. However, there is no evaluation during the multiple steps in the dataset creation. It would be very helpful if the authors could evaluate a sample of instances during each step in the creation, for example, the accuracy of entity linking, and also provide case studies on the errors. Because some steps involve rule-based approaches or RegEx, to be more confident about the quality of the dataset, proper evaluation during the creation process will be helpful.

A related issue is: how to make corrections to the dataset. There are errors noted in evaluating the 500 samples. The question here is: what is the best way to correct some of these errors once the data is related to the research community.


**Additional Feedback:**

The authors do not give much discussion on the results reported in Table 4. What is worrying here is the significant differences between the absolute values of the measures across different models on the same dataset.

The authors have well addressed most of my comments through feedback.

**Clarity:**

The paper is well written in general. To improve, the Introduction section can be better organized. The few paragraphs in the Introduction can be better arranged to define event, provide the background of event-centric KB, and the current status of the KB to text generation. It reads to me that the authors emphasize the dataset is largest for multiple times, but do not discuss much on the quality of the dataset.



**Correctness:**

The overall creation process is reasonable. Nevertheless, the authors do not present the detailed creation process in a more formal way (e.g., the entity matching algorithm).

**Documentation:**

The dataset creation process is reasonably well documented. The writing can be more formal in Section 2. Examples shall be given to illustrate the process of the data linking and refinement between EventKG and Wikipedia. In Section 3.3, it is not reported that each annotator evaluates the 500 samples, all the 500 samples are shared among the annotators for evaluation. Is there disagreement among the annotators?

Authors have answered this question through feedback and the revised version shall be clearer on this point.

**Ethics:**

There is no ethical concern.

**Relation To Prior Work:**

The authors did a good job to position this dataset with existing ones (to my best knowledge).

**Summary And Contributions:**

In this resource paper, the authors provide the EventNarrative dataset for event-centric knowledge graph to text generation. The dataset was created by careful linking (and refining) the elements in EventKG and Wikipedia articles.  The main contributions include the EventNarrative dataset itself, the procedure for the linking and refining, and an evaluation of the baseline models on this dataset.

---

> ### Author Response · Authors · 2021-07-10
> **Addressing comments part 1**
>
> We sincerely appreciate the thoroughness that went into your review. Thank you for all of the constructive comments. We do our best to address and improve on your comments below:
>
> >it is not clear the unique characteristics of event-centric KG compared to generic KB with respect to the task of KG to text generation. The authors may want to provide a stronger justification on why such a dataset is essential to study KG to text generation
>
> Thank you for bringing this to our attention. We will clarify the differences between event-centric KG compared to a generic KB with respect to text generation. We will also paraphrase the paper and add the following to provide a stronger justification on why such a dataset is essential to study KG to text generation:
>
> “Event-based graphs are more in need of narration, that is because it is easier for humans to digest them in natural language rather than their graph format due to the complex structure and content. Furthermore the process of extracting event-centric entities is weaker, so event centric KGs have different missing data distributions and our dataset can facilitate training specific text-graph or graph-text generation models. Also, our event-centric KG has a richer ontology (see Table 2) than previous KGs. “
>
> >Another weakness is on the quality of the dataset creation. The authors evaluated 500 randomly sampled KG-text pairs and reported a very good score. However, there is no evaluation during the multiple steps in the dataset creation. It would be very helpful if the authors could evaluate a sample of instances during each step in the creation, for example, the accuracy of entity linking, and also provide case studies on the errors. Because some steps involve rule-based approaches or RegEx, to be more confident about the quality of the dataset, proper evaluation during the creation process will be helpful.
>
> We agree that more fine grain evaluation during the data creation process will be helpful. Therefore, we had designed and constructed our qualitative analysis in order to do so (we will try to emphasis/clarify this in the paper). During our manual evaluation we asked annotators to carefully evaluate all entities, which relates to the entity matching step; and relations/triples which relates to the narrative KG generation step. Annotators also reported errors which were mischaracterized by Wikipedia as well as dates which have different scopes, which relate to both the sources and our RegEx matching steps.
>
> >A related issue is: how to make corrections to the dataset. There are errors noted in evaluating the 500 samples. The question here is: what is the best way to correct some of these errors once the data is related to the research community.
>
> Thank you for posing this question. We plan to make corrections/updates to the dataset, which we will announce on our website/GitHub. Please see the DataDocumenation.pdf file, section A.7 Maintenance for more details. As you are probably concerned too, this might cause confusions comparing future model’s performances, and to mitigate that we will make sure to clearly state any big changes to the dataset. We also distribute our dataset, which can be used for other tasks that may correct the errors in the dataset.
>
> >The overall creation process is reasonable. Nevertheless, the authors do not present the detailed creation process in a more formal way (e.g., the entity matching algorithm).
>
> Thank you for pointing this out. We hope to clarify this. In our dataset creation process we do not use any existing libraries for entity matching, as they did not perform well on event-centric data. Please see section 3.4 Limitations.
>
> >To improve, the Introduction section can be better organized. The few paragraphs in the Introduction can be better arranged to define event, provide the background of event-centric KB, and the current status of the KB to text generation. It reads to me that the authors emphasize the dataset is largest for multiple times, but do not discuss much on the quality of the dataset.
>
> We will do our best to more clearly define event and provide some more background of event-centric KB so that the introduction can be more easily followed. Further, we can add to the introduction about the quality of our dataset and point to the qualitative analysis.
>
> >The dataset creation process is reasonably well documented. The writing can be more formal in Section 2. Examples shall be given to illustrate the process of the data linking and refinement between EventKG and Wikipedia.
>
> We want to make our data creation process as clear as possible. Therefore, we will add another more detailed flow chart (similar to Figure 2) in the appendix which will detail each step of the process. We would be grateful if you remember any specific pain point when reading those sections so we can make sure to address them.

---

> > ### Comment · Reviewer_kKsU · 2021-07-15
> > **Thank you for the feedback**
> >
> > Thank you for addressing my comments. Overall, I believe we are working together to improve this work and make it more impactful and useful to the community.
> >
> > In general, I agree with the clarifications and the planed "future work" to be implemented. However, I still find that the justification for "event-centric KB" is not strong. We do not see the uniqueness of event-centric KB and why text generation models need to be specifically trained for event-centric KB (or event-centric KB need to be processed differently from other kind of KBs). A related question is: given a graph-to-text task, is there a need to first recognize a give graph instance is an event-KB before calling the generation model?

---

> > > ### Comment · Program_Chairs · 2021-07-19
> > > **Additional comment**
> > >
> > > Thank you for your response, we hope to clear up your concern and appreciate your questions as it encourages exciting discussion on event-centric KBs. We agree that on high-level, event or entity centric KBs don’t necessarily need to be treated differently for the tasks of conditional text generation, KB to Text or Text to KB.
> > >
> > > However, event centric and entity centric KBs are from two different domains with different statistical properties. For example, each has a different degree distribution: Event centric KBs usually have nodes representing the ‘event’ which have a high degree and many different relation types. While nodes in entity-centric KBs might have a huge number of edges, but often very few unique relations (e.g. ‘NYC’ in YAGO has a very high degree, but all the edges are from a small subset of the ontology such as located_in and lives_in. While ‘Battle of Stalingrad’ from Wikidata, has many different relations describing time, location, actors, target, etc).
> > >
> > > The question is whether this difference will cause problems for models designed/trained for one and applied to another. Which is one of the questions we hope our dataset will help in understanding. If the answer is yes (which our preliminary experiments with current graph to text models show this is likely the case), we hope it will help develop better models suitable for each or designing models that can work with both.
> > >
> > > If the answer is “current models can handle both KB types the same”, we hope the significantly larger size and richer ontology of our dataset will help train and improve these models in general.

---

> ### Author Response · Authors · 2021-07-10
> **Addressing comments part 2**
>
> >In Section 3.3, it is not reported that each annotator evaluates the 500 samples, all the 500 samples are shared among the annotators for evaluation. Is there disagreement among the annotators?
>
> Yes, all of the same samples are shared among the annotators. We will calculate the kappa agreement and report here and in the paper soon.
>
> >The authors do not give much discussion on the results reported in Table 4. What is worrying here is the significant differences between the absolute values of the measures across different models on the same dataset.
>
> This is an interesting observation that has also been seen in previous work, e.g. Ribeiro et al. 2020 [1] also show similar scales of differences in performance when evaluating on 3 different datasets with similar measures. This could be partially explained by the expressive powers of the baselines used (e.g. number of params, pre-training, and their architects are significantly different) but we hope the existence of such datasets as ours can help further study and explain these phenomena.
> [1] https://arxiv.org/pdf/2007.08426.pdf

---

### Official Review · Reviewer_ipe9 · 2021-07-05
**Review on - EventNarrative: A large-scale Event-centric Dataset for Knowledge Graph-to-Text Generation**

**Rating:** 7
**Confidence:** 4

**Strengths:**

The proposed dataset - 'EventNarrative' is 6 times larger than the current largest parallel dataset. It covers events across a wise range of domains and time periods. The knowledge graph-to-text pairs are generated automatically using a rich ontology, which paired with a its large scale nature, would form dense and information graphs, The qualitative analysis conducted, with the help of 3 human annotators, on 500 randomly sampled Knowledge graph-to-text pairs showed good accuracy values - ~96% for entities and 95% for relations.
The authors have provided a comparison of the dataset across other popular datasets - WebNLG, AGENDA, GenWiki (full and fine). WebNLG and EventNarrative, both are parallel, have all entities from the knowledge graph present in the narrative and all entities are linked to the graph triples. However, EventNarrative is a significantly larger dataset. Additionally, as EventNarrative is sourced from wikipedia, various existing tools can be used on this dataset. This will be beneficial for the broader research community to carry out further experiments and evaluate models in the domain.


**Weaknesses:**

In the dataset creation, there are several limitations which have been highlighted by the authors in section 2.4.
These talk about missing out on entities found within the narrative test, due to incompleteness of wikidata, which is used as a key source in the data creation.
They also highlight the poor performance of co reference resolution systems in event centric data.
In addition to these, it is not clear if any sort of semantic similarity is checked for the events or properties in this process. If only an exact match is being considered, it may lead to missing or double counting of certain events or properties.

In section 3.3, the qualitative analysis is described.
Here, 500 randomly sampled KG to text pairs are chosen for manual checking. As there are 230k such pairs in the data, 500 seems to be a small sample size. The results produced in this analysis, may be biased and will possibly not be a correct representation of the entire dataset.

In section 4.1, the authors describe the experimental setup for the benchmark evaluation. This is done in detail and will definitely help in reproducing the results. However, there is no insight on the reasoning behind choosing the particular parameters mentioned.

**Additional Feedback:**

1. The AGENDA dataset is first mentioned in line 48. The source for this should be cited here. It has been cited in later occurrences (line 174 and later), but the first mention should have the relevant citation.
2. Section 5 can be moved up in the structure of the paper. Section 5 describes the related datasets which have been used for comparisons and analyses all throughout section 3 and 4. It makes sense to introduce the datasets before these sections.
3. In section 2, there are a lot of steps which highlight the data creation process. It becomes a little difficult to follow as there are multiple steps. It would be very helpful if you could provide a relevant example to help explain the process.

4.In section 3.3, the qualitative analysis is described. It'll be helpful if you could provide more detail on this process. Why was a sample size of 500 pairs chosen? Were there any steps taken to ascertain no or low bias in this data? Why do you think this sample will be a good representation of a dataset with 230,000 pairs?

5. In section 4.1, the experimental setup for the benchmark evaluation. The paper provides the specific details of the parameters and models chosen. How were these parameters decided? Did you conduct experiments with other configurations? Can you share the comparison details for these or any explanation on why the mentioned configuration is preferred?

**Clarity:**

There are some subsections which are a little hard to follow as there are a lot of details present - sections 2.1 and 2.2. These highlight the initial stages of the dataset creation process. There are a lot of steps which have been condensed into paragraphs in these sections. If they could be broken down into bullet points or restructured, it may be easier to follow.
A real example from the dataset, for these steps, would also help the reader understand better, in addition to the flowchart already provided (Figure 2).

Also,  the section on Related Datasets (Section 5) should be moved up. This will help build context on these datasets which have been used for comparisons throughout sections 3 and 4.

**Correctness:**

The claims in the submission seem to be correct and appropriate references and comparisons have been provided with existing work. There is also qualitative analysis done to ascertain the quality of the dataset which has given a good score, in the selected sample dataset.
The dataset creation process is conducted in a thorough manner and the steps have been described in detail. For each step, the necessary steps were taken as needed. - eg. addition of the date module in keeping with the wikipedia style manual etc. The reasonings and explanation for these were given as well. - Section 2.
In section 4, the benchmark evaluation process is described. This is done in 2 steps:
1. modeling the problem with a graph transformer-based network - GraphWriter
2. treating the problem as a summarisation task by fine-tuning on pre-trained language models - BART and T5



**Documentation:**

The paper describes the data creation process in detail in section 2.
A kaggle link to the dataset is present, along with the hosting, licensing, and maintenance plan in the supplementary material.
In section 7, possible broader impact of the dataset is described. Risks concerning fake news and disinformation pertaining to recent events showing up in the dataset has been mentioned. The authors clearly highlight this and discourage anyone from substituting any text which may spread disinformation.

**Ethics:**

No, to the best of my knowledge, there don't seem to be any ethical concerns which need further review.

**Relation To Prior Work:**

The authors have provided a comparison of EventNarrative with other popular datasets - WebNLG, AGENDA, GenWiki (full and fine) in section 3. Section 5 talks about all the mentioned datasets. The comparison is done on the following fronts: total number of knowledge graph components, number of tokens in the narratives, overall mean statistics, domain, ontology etc. Table 1 and Table 2 highlight these in detail.
The paper states, on the basis of these analyses, that "EventNarrative is the only dataset that is large-scale, ontology-based, utilizes an open real-world KG, may be used for supervised learning, and contains fully connected graphs that are fully contained within a text narrative"

**Summary And Contributions:**

The paper introduces ‘EventNarrative’ : an event centric knowledge graph to text dataset, which is about 6 times larger than the current largest parallel dataset. The paper highlights the data creation process in detail and provides a final synopsis of the generated EventNarrative dataset. The sources used are EventKG along with wikidata, enhanced with full text from wikipedia. This is followed by entity matching and Narrative graph generation after various methods of filtering of the events, and a separate module to capture dates. The final dataset is then compared compared with existing datasets by size and data sparsity. There is also a qualitative analysis conducted, using human annotators, to ascertain the quality of the data. Further, a benchmark evaluation is also done.

 The main contributions of the paper are as follows:
1. A large-scale, event-centric, parallel knowledge graph-to-text dataset of over 230,000 KG to text pairs, spanning over 7,000 event types, and over 650,000 triples
2. A comprehensive entity matching and knowledge graph-to-text matching algorithm that automatically pairs KGs to natural language texts
3.  Benchmark evaluations and baselines results on EventNarrative

---

> ### Author Response · Authors · 2021-07-09
> **Our Thanks and Reply to Reviewer 1**
>
> We sincerely appreciate the thoroughness and detail that went into your review. Thank you for all of the constructive comments. We will first address those comments that we can respond to right now, to facilitate and start the conversation so we can better improve the paper based on the feedback. However, please be assured that we will work to address/incorporate all of the feedback. For ease of reading, we have noted the comments in block quotes and their responses below:
>
> 1.
> > In section 3.3, the qualitative analysis is described. Here, 500 randomly sampled KG to text pairs are chosen for manual checking. As there are 230k such pairs in the data, 500 seems to be a small sample size. The results produced in this analysis, may be biased and will possibly not be a correct representation of the entire dataset.
> >In section 3.3, the qualitative analysis is described. It'll be helpful if you could provide more detail on this process. Why was a sample size of 500 pairs chosen? Were there any steps taken to ascertain no or low bias in this data? Why do you think this sample will be a good representation of a dataset with 230,000 pairs?
>
> Thank you for bringing up this point. We will also report confidence intervals of our error rates based on our sample size (500 iid samples) to the paper. For quicker reference, the 95% confidence interval for the error rates (CE and RC assuming both have 0.045) is [0.025, 0.065].
>
> 2.
> >In section 4.1, the authors describe the experimental setup for the benchmark evaluation. This is done in detail and will definitely help in reproducing the results. However, there is no insight on the reasoning behind choosing the particular parameters mentioned.
> >In section 4.1, the experimental setup for the benchmark evaluation. The paper provides the specific details of the parameters and models chosen. How were these parameters decided? Did you conduct experiments with other configurations? Can you share the comparison details for these or any explanation on why the mentioned configuration is preferred?
>
> We agree that this issue is important to address. Given the computational complexity and the  environmental and economical cost (3-5 days to complete T5 and BART based models) of experimenting multiple hyperparameters for these models, we used a set of sensible params ( used in previous work from Ribeiro et al. (2020) and Wolf et al. (2019)).
>
>
>
> 3.
> >There are some subsections which are a little hard to follow as there are a lot of details present - sections 2.1 and 2.2. These highlight the initial stages of the dataset creation process. There are a lot of steps which have been condensed into paragraphs in these sections. If they could be broken down into bullet points or restructured, it may be easier to follow. A real example from the dataset, for these steps, would also help the reader understand better, in addition to the flowchart already provided (Figure 2).
> >In section 2, there are a lot of steps which highlight the data creation process. It becomes a little difficult to follow as there are multiple steps. It would be very helpful if you could provide a relevant example to help explain the process.
>
> We want to make our data creation process as clear as possible. Therefore, we will add a more detailed flow chart (similar to Figure 2) in the appendix which will detail each step of the process. We would be grateful if you remember any specific pain point when reading those sections so we can make sure to address them.
>
> 4.
> >Also, the section on Related Datasets (Section 5) should be moved up. This will help build context on these datasets which have been used for comparisons throughout sections 3 and 4.
> >Section 5 can be moved up in the structure of the paper. Section 5 describes the related datasets which have been used for comparisons and analyses all throughout section 3 and 4. It makes sense to introduce the datasets before these sections.
>
> Thank you, we agree that this helps the flow of the paper, we have moved the section.
>
> 5.
> >A link to the dataset URL is present, however, there seems to be no hosting, licensing or maintenance plan.
>
> We have addressed these in DataDocumentation.pdf in the supplementary material for the hosting, licensing, and maintenance plan. Specifically, please see A.5, A.6 and A.7.
>
> 6.
> >The AGENDA dataset is first mentioned in line 48. The source for this should be cited here. It has been cited in later occurrences (line 174 and later), but the first mention should have the relevant citation.
>
> Thank you for catching this. We have made this correction.

---

> > ### Comment · Reviewer_ipe9 · 2021-07-20
> > **Thank you for addressing my comments**
> >
> > Thank you for addressing all my comments and concerns.
> > Points 1 and 2 provide more clarity now and it will be great if these details are included in the paper.
> > For point 3, an example which goes through the entire process of data creation will largely address this concern. A detailed flow chart, as mentioned, should be helpful here.

---

### Decision · Program_Chairs · 2021-07-26

**Decision:**

Accept

**Comment:**

In this paper the authors provide a new very large dataset of 230k graphs for knowledge graph to text generation.  All reviewers found the large dataset itself and the process by which it was generated from reliable sources to be an important contribution with significant benefit over prior work.